# Mindfulness-Based Intervention for the Reduction of Compassion Fatigue and Burnout in Nurse Caregivers of Institutionalized Older Persons with Dementia: A Randomized Controlled Trial

**DOI:** 10.3390/ijerph191811441

**Published:** 2022-09-11

**Authors:** Victoria Pérez, Ernesto J. Menéndez-Crispín, Carmen Sarabia-Cobo, Pablo de Lorena, Angela Fernández-Rodríguez, Julia González-Vaca

**Affiliations:** 1Institute CR Santa Lucía, 28001 Madrid, Spain; 2Centro Interdisdiplinar de Psicoterapia, 28001 Madrid, Spain; 3Facultad de Enfermería, IDIVAL, Universidad de Cantabria, Avda Valdecilla s/n, 39011 Santander, Spain; 4CAD Santander, IDIVAL, 39008 Santander, Spain; 5Nursing Research Group (GRIN) from the IDIBELL Translational Medicine Area, University of Barcelona, 08007 Barcelona, Spain

**Keywords:** mindfulness-based intervention, nurse, dementia, burnout, compassion fatigue, occupational stress

## Abstract

The recent COVID-19 pandemic has severely impacted the mental health of nurses caring for institutionalized older people. Caring in this environment can be complex, with higher levels of burnout and compassion fatigue in staff. It is therefore important to find interventions to increase the well-being of staff. Mindfulness training is known to be effective in treating a variety of physical and mental health conditions. This study sought to conduct a direct evaluation of the effectiveness of a combined online training in two types of mindfulness-based therapies for the reduction of compassion fatigue and burnout in geriatric nurses caring for institutionalized elderly people with dementia. In a randomized controlled trial (*n* = 39 experimental group, *n* = 35 control group), we explored whether individuals with high levels of burnout and compassion fatigue would benefit more from an online mindfulness training program. The outcome variable was the ProQoL professional quality of life scale, which was collected at baseline, at six weeks, and at three months after completion of the intervention. Significant differences were found between both groups for the subscales Compassion Fatigue and Burnout (*p* < 0.05), with a significant improvement in the experimental group (significant effect size). These findings were maintained at three months after the end of the intervention for both compassion fatigue (F1,28 = 18.14, *p* = 0.003) and burnout (F1,28 = 7.25, *p* = 0.040). However, there were no differences between groups for the satisfaction subscale. The effect of time and the effects of comparing the two groups after controlling for time were statistically significant for all three subscales of the questionnaire (all *p* values < 0.001), with effect sizes ranging from small to large (R^2^ change 0.10–0.47). These data indicate that the experimental condition was more effective, explaining between 10 and 18% more of the variance. A short, online intervention based on mindfulness training appears to be effective for reducing compassion fatigue and burnout in geriatric nurses, with sustained effects over time.

## 1. Introduction

Levels of occupational stress or burnout are higher in health professionals than in other workers [1]. The negative consequences of not having adequate stress-coping strategies to deal with the demands of professional life have an impact on the health and mental well-being of the individual, as well as on their professional performance [2]. This situation has clearly worsened with the COVID-19 pandemic [3].

Staff providing long-term hospital care to older and geriatric patients are exposed to numerous factors that can lead to the development of burnout syndrome [4]. Burnout is associated with an increased risk of absence from work, low job satisfaction, and an increased intention to quit [5]. Considering that the number of geriatric nursing staff is already insufficient, research is needed on interventions aimed at reducing work-related stress in inpatient care for older people [6].

Caring for patients with dementia generates significant job stress that can result in employee dissatisfaction and mental exhaustion [7]. Part of the stress stems from burnout, the chronic psychological syndrome of perceived job demands exceeding perceived resources in the work environment [4].

### 1.1. Theoretical Framework

The Cognitive Activation Theory of Stress [8] (CATS) defines stress neutrally, meaning that stress in itself is neither good nor bad. However, Ursin and Eriksen [8] argue that prolonged stress and/or excessive stress loads can be detrimental to the individual. The starting theoretical framework is that staff working in nursing homes exhibit high levels of compassion fatigue and burnout, marked by the characteristics of their work, as the main elements of stress [9]. Many factors elevate caregiver stress, including the nature and severity of the challenging behaviors of dementia patients, working conditions, degree of support from staff management, working conditions, work overload, etc.

#### 1.1.1. Compassion Fatigue

Researchers have also shown that compassion fatigue can affect the professional caregiver as well as the workplace, causing decreased productivity, a greater number of sick days, and increased turnover [10]. However, few validated reports have detailed the incidence and prevalence of compassion fatigue in psychogeriatric nurses caring for older people with dementia [5].

The condition of compassion fatigue was first identified by Joinson [11] in a study of burnout in nurses working in an emergency department. The researcher identified characteristic behaviors of compassion fatigue, including chronic fatigue, irritability, fear of work, aggravation of physical ailments, and lack of joy in life. Later, Figley [12] defined compassion fatigue as a state of tension and preoccupation with clients’ individual or cumulative traumas. The phenomenon of compassion fatigue arises suddenly and without warning and includes a sense of helplessness and confusion. Figley has described it as the toll a caregiver experiences as a result of caring for others. Compassion fatigue results from giving high levels of energy and compassion over a prolonged period to those who are suffering, often without experiencing the positive outcomes of seeing patients improve [12]. The term compassion fatigue has been applied to a disconnection or lack of empathy on behalf of professional caregivers. Empathy and emotional investment have been found to potentially cost caregivers, putting them at risk [13]. Compassion fatigue has been equated with burnout, secondary traumatic stress disorder, vicarious traumatization, secondary victimization or co-victimization, compassion stress, emotional contagion, and countertransference. Numerous studies exist on nurses’ experiences of compassion fatigue and burnout [10,13,14,15]. Most nurses who self-identified as having compassion fatigue described a change in their practice whereby they began to protect and distance themselves from the suffering of patients and their families [14,15]. Feelings of irritability, anger, and negativity emerged, although participants described denying or ignoring these emotions to try to overcome their workday. Difficulties with work carried over into the nurses’ personal lives, affecting their relationships with family and friends [15]. Such experiences invariably challenged the participants’ identity, making them reflect on the type of nurse they were. Participants’ compassion fatigue created a sense of hopelessness regarding positive change, although some nurses described strategies that seemed to help alleviate their compassion fatigue.

Coping strategies can reduce nurse burnout and maintain effectiveness for six months to one year [16]. Clearly, there is an increasing demand for wellness-support methods at work [17]. Interventions based on mindfulness are gaining special relevance in recent years, as they have the potential to improve the psychological wellbeing of nurses, a result of the research carried out over the last 20 years in this regard [18]. Now, with the situation generated by the COVID-19 pandemic, these interventions are more necessary than ever [19].

#### 1.1.2. Mindfulness-Based Interventions (MBIs)

Mindfulness-based interventions have experienced a remarkable increase in scientific and popular interest over the last two decades [20,21]. Mindfulness can be defined as an approach to experiencing everyday life by directing attention and awareness to the present moment without judgment. Mindfulness encompasses the key therapeutic concepts of acceptance, compassion, and detachment [22]. As a therapy, mindfulness practice is predominantly based on a program that was originally established for mindfulness-based stress reduction (MBSR). Since the 1979 debut of Jon Kabat Zinn’s Mindfulness-Based Stress Reduction (MBSR) program [23,24], its structure and form have become the model for subsequent mindfulness-based programs (MBP). A notable example is the equally popular Mindfulness-Based Cognitive Therapy (MBCT) program developed two decades ago [25,26]. Common features of these programs include (i) a standardized, instructor-led experiential learning routine of about 2 to 2.5 h each week, (ii) a day of silent mindfulness practice around the sixth week, and (iii) daily formal and informal practices assigned each week that (iv) require approximately 45 min of daily dedication for (v) six days a week. Admittedly, some programs may have a longer duration, such as Mindful Self-Compassion (MSC), which lasts ten weeks [24], or the relatively simpler and more abbreviated approaches such as the Finding Peace in a Frantic World program [27] developed for companies or school settings, which usually lasts six weeks.

Sufficient evidence has accumulated over the last twenty years to indicate MBI’s efficacy in addressing anxiety, burnout, and stress problems in both clinical and professional populations [28]. MBIs are effective at improving many biopsychosocial conditions, such as depression, anxiety, stress, insomnia, addiction, psychosis, pain, hypertension, weight management, cancer-related symptoms, and prosocial behaviors [29,30]. MBIs have been found to be beneficial in healthcare settings, in schools, and in the workplace; however, further research is warranted to examine its efficacy in relation to different problems [28]. A literature review included a total of 142 non-overlapping samples and 12,005 participants [31]. In posttreatment, mindfulness-based interventions were superior to no treatment, minimal treatment, non-specific active controls, and specific active controls. Mindfulness conditions did not differ from evidence-based treatments. The results of meta-analyses supported the notion that there was evidence supporting mindfulness-based interventions as a treatment for disorders associated with depression, anxiety, and stress.

In another systematic review, which focused on studies measuring the impact of mindfulness-based interventions on physicians’ well-being and performance, the findings suggested that physicians benefited positively from greater mental well-being after mindfulness-based interventions [32].

Another recent systematic review included 85 randomized controlled trials [33], 79 of which reported significant positive effects on at least one health-related outcome, and over a quarter of these targeted a clinical population. Most studies focused on psychological outcomes, such as reduced anxiety and depression, as well as emotional regulation, stress, and cognitive outcomes. These were found in brief programs as short as 5 min.

In response to questions about the scientific basis of interventions based on MBI, a recent investigation assessed their empirical status by systematically reviewing meta-analyses of randomized controlled trials (RCTs) [34]. A total of 160 effect sizes were reported in 44 meta-analyses (k = 336 RCTs, N = 30,483 participants). Intervention groups based on the use of MBI showed superiority to control groups in most studies. The effects of MBIs were similar or superior to other intervention groups using conventional or evidence-based treatments, especially for stress management.

Mindfulness has therefore been shown to improve patient care and reduce work stress and nursing staff turnover [35,36]. Improved awareness and the use of mindfulness strategies have the potential to improve patient outcomes, reduce the cost of professional turnover, and improve patients’ emotional well-being and job quality [37].

Therefore, we considered it important to carry out an effective, measurable, replicable, and simple intervention to promote emotional self-care measures in nurses working with older people with dementia in the current pandemic context. The aim of the present study was to conduct a direct evaluation of the efficacy of a combined online training in two types of mindfulness-based therapy for the reduction of compassion fatigue and burnout in geriatric nurses caring for institutionalized older people with dementia in a randomized controlled trial.

## 2. Materials and Methods

This study used a randomized controlled trial with a control group and an intervention group.

### 2.1. Participants

Purposive sampling was used, based on a total sample of 82 nurses who were initially recruited from twelve elderly care centers belonging to the same religious foundation in six cities of Spain. A total of 74 nurses agreed to participate. The characteristics of the nursing homes were similar in terms of the number of elderly people cared for, their degree of dependency, staffing, and work shift characteristics. The participants who agreed to participate in the study were randomly assigned to the experimental condition (*n* = 39) and to the control condition (*n* = 35), with the reference that they had to belong to the same center. Thus, six centers were considered the intervention group and six centers were considered the control group to avoid contamination of the sample. There were no dropouts, and all participants completed the study. The flow of participants through each stage of the trial is represented in the CONSORT diagram (Figure 1).

The intervention was carried out between September and February 2021.

### 2.2. Variables

Sociodemographic variables such as age, sex, marital status, years of dedication to the field of geriatrics, and hours of work per week were collected.

#### 2.2.1. Dependent Variable

The dependent variable was the Professional Quality of Life Scale (ProQoL) in its Spanish adaptation [38,39]. This questionnaire contains 30 items that measure three constructs: compassion satisfaction, burnout, and compassion fatigue [40]. Compassion satisfaction is defined as pleasure derived from being able to do one’s job well; burnout is defined as feelings of hopelessness and difficulty coping with work or doing a job effectively, as well as trauma. Compassion fatigue is defined as secondary work-related exposure to extremely stressful events. The scale has been shown to have good psychometric properties and has been validated in different countries [41]. In this study, the α-reliability of each subscale was 0.88, 0.76, and 0.81, respectively. Response options ranged from 0 (never) to 5 (always). For scoring, five items belonging to the burnout subscale must be reversed. There is no total score; the score for each construct is obtained from the sum of the values of the 10 items of each subscale. Scores are interpreted as low (≤22), average (23–41), and high (≥42).

#### 2.2.2. Independent Variables: Training Program

Participants were randomly assigned to one of two groups: (a) a six-week mindfulness training program delivered through an online platform or (b) a three-month wait-list control after completion of the intervention in the experimental group.

### 2.3. Outcome Measures

The ProQOL R-IV scale was collected at baseline, at six weeks of intervention (T1), and at three months after the end of the intervention (T2).

### 2.4. Intervention

The intervention was a six-week group intervention based on the principles of Mindfulness-Based Stress Reduction [23,24,42]. Kabat Zinn’s research has had a great impact on the development of mindfulness programs applied to different settings, with solid evidence regarding their effectiveness [43,44]. Our intervention was based on his teachings and combines elements of MBCT [45]. The intervention was designed to decrease or prevent compassion fatigue, to help learn to manage stressful situations, and to increase compassion satisfaction, based on the literature and the training of the researchers who designed it [16,46].

The main characteristics were: six recorded sessions of 60 min each, with videos and interactive exercises led by a nurse and a psychologist trained in mindfulness. All sessions followed the same structure: they began with a brief relaxation and breathing technique, continued with the content of the session, and ended with a quote for personal reflection on the topic covered and an individual reflective writing exercise. In addition to the sessions, twelve assignments were also made available on the platform for participants to practice in daily life with supporting emails (two per session). There were also five guided meditation audio downloads. Participants could pause the course and repeat any part at any time. There was also an email address and phone number for general and technical support.

The entire course was hosted on a free platform (Moodle), which each participant accessed with a username and password. Each week, the session was made available (on Mondays) along with the two tasks for the week, and every two weeks, an audio-guided meditation was provided. The platform informed the user of their progress with notifications.

### 2.5. Procedure

Participants were randomly assigned to one of two groups: (a) a six-week training program or (b) a three-month wait-list control after completion of the intervention.

After signing the informed consent form, participants received a link by e-mail to self-complete the study variables at three points in time: baseline, at six weeks (T1), and at three months after completing the intervention (T2).

### 2.6. Ethical Considerations

All procedures were in accordance with the 1975 Declaration of Helsinki, revised in 2000. All participants gave written informed consent. The study was approved by the ethics committee of the foundation (EC 09/2021) and the managements of the 12 participating centers. All data were anonymized and treated according to the current legislation of the country.

### 2.7. Statistical Analysis

IBM SPSS Statistics v.24 n.0. (INTERNATIONAL BUSINESS MACHINES CORPORATION, Armonk, NY, USA) software was used for the statistical analysis. A bilateral contrast and a 95% confidence level were adopted. A descriptive analysis of all the variables collected was performed for each group. Possible differences between the baseline characteristics of the study groups were evaluated with t-tests for continuous variables and χ2 tests for categorical variables. To examine differences in outcomes between the intervention and control groups during the intervention period, a repeated-measures ANOVA analysis of variance was performed to analyze differences between the experimental and control groups between baseline and T1 and T2. Another repeated-measures ANOVA was performed for the intervention group to test for intrasubject effects between baseline and T1 and T2. The significance level was set at *p* < 0.05. Change over time within each condition was examined by treating each group as an n of 1. Multiple linear regression (MLR) was used to estimate the extent to which subject variables were predicted by the type of intervention (experimental vs. control) after controlling for the effect of time. Effect size in the regression analysis is reflected by R2 (which is the percentage of variance explained by the linear relationship between two variables), and specific effects were estimated by the change in R2, which is interpreted as R2 of 0.02 = small, 0.15 = medium, and 0.26 = large. All study variables met the principles for applying MLR: skewness and kurtosis values within ± 1, no significant outliers, and no evidence of multicollinearity (variance inflation factor <5).

## 3. Results

The total sample analyzed consisted of 74 nurses (89.6% female and 10.4% male), with a mean age of 37 years (SD = 9.13), and an age range of 25 to 56 years. Most of the sample was married (70.2%), 8 were single, 19 were divorced, and 13 were domestic partners. The mean years in practice were 11.52 (SD = 9.25), and most of the nurses worked 40 h per week (79.8%).

Demographic characteristics of the participants in both groups were compared by a series of chi-square and independent samples t-tests, which indicated no statistically significant differences between the groups (*p* > 0.05). All subjects in the experimental group completed the entire training program.

Table 1 shows the values for the baseline phase of both groups in the ProQoL questionnaire. There were no statistically significant differences in the variables between the groups at baseline, nor were significant correlations established between the sociodemographic variables and the subscales.

To measure the effectiveness of the intervention, a repeated-measures ANOVA was performed. At six weeks, immediately after the intervention, the level of compassion fatigue decreased significantly in the experimental group (M 9.21, SD 11.24) compared to the control group (M 18.23, SD 6.23) (F1,65 = 8.15, *p* = 0.011). The same phenomenon occurred in the burnout subscale (F1,65 = 11.05, *p* = 0.02), with scores of 11.23 (SD 6.10) for the experimental group and 19.21 (SD 8.32) for the control group. However, satisfaction values remained similar for both groups, with no statistically significant differences.

Longitudinal measurement at three months after completion of the intervention showed that the initial decrease in the level of compassion fatigue remained significant (F1,28 = 18.14, *p* = 0.003), which was also the case for the burnout subscale (F1,28 = 7.25, *p* = 0.040). Levels of compassion satisfaction remained similar to the baseline for both groups, with no statistically significant differences.

The effect size also increased from the baseline to the final phase (T2) for both the compassion fatigue subscale (baseline Cohen’s d = 0.32; T2 Cohen’s d = 2.34) and the burnout subscale (baseline Cohen’s d = 0.12; T2 Cohen’s d = 2.48). However, it remained stable for the satisfaction subscale.

In Table 2, we can see the multiple linear regression analyses predicting subject variables, including the three subscales (ProQOL). The effect of time and the effects of comparing the two groups after controlling for time were statistically significant for all three subscales of the questionnaire (all p’s < 0.001), with effect sizes ranging from small to large (R^2^ change 0.10–0.47). These data indicate that the experimental condition was more effective, explaining between 10 and 18% of the variance in the psychological data, with all coefficients being β-significant and greater than 0.31, reflecting substantial psychological improvement in the experimental subjects.

## 4. Discussion

The present study aimed to conduct a direct evaluation of the efficacy of a combined online training in two types of mindfulness-based therapies for the reduction of compassion fatigue and burnout in geriatric nurses caring for institutionalized older people with dementia in a randomized controlled trial. Very few studies have investigated effective ways to reduce stress in staff caring for people with dementia in nursing homes in relation to compassion fatigue and caregiving satisfaction during this period [47,48,49]. This trial will extend our knowledge by evaluating whether online mindfulness training reduces stress and improves job satisfaction in this professional group.

In view of the high rate of completion and attendance, the findings of our study suggest that our combined intervention, based on MBSR and MBCT, is a feasible and acceptable psychosocial program for the target population. The results obtained in achieving the main objective show that online mindfulness-based training effectively decreased levels of compassion fatigue and burnout up to three months after the end of the intervention compared to a control group with similar characteristics. Although such a conclusion may be somewhat premature given the heterogeneity of mindfulness-based interventions and the lack of independent research groups replicating specific mindfulness-based interventions, the field clearly indicates that engaging in mindfulness-based practices helps caregivers improve their well-being [18,50,51]. Furthermore, when caregivers engage in mindfulness-based practices, they improve their clients’ quality of life by reducing or eliminating the use of restrictive procedures, such as physical restraints and emergency psychotropic medications [52].

Our findings suggest that these processes mediate some of the effects of MBI on nurses’ psychological functioning. We found that changes in compassion fatigue and burnout were reasonably predicted by the intervention, although satisfaction with care remained stable. In addition, the passage of time was also a modulating variable for intervention effectiveness. Our findings are similar to other studies that employed similar interventions and conducted a longitudinal follow-up [53,54,55]. Baseline levels of compassion fatigue and stress in both groups are average values, comparable to those found in other similar studies [5,10,56]. In a recent and interesting study in Italian healthcare professionals evaluating baseline stress levels before and after the pandemic and the effect of Mindfulness-Based Stress Reduction (MBSR) training on well-being (PGWBI), stress (PSS), and burnout (MBI), the authors concluded that MBSR training may represent an effective strategy to reduce distress in an emergency period [57]. Although the type of intervention used in this study had a different approach, the results are similar to our study. Although we did not specifically look at stress levels before and after the pandemic, we did conduct the intervention during the pandemic. The stress levels are similar to those achieved in the Italian study, although the latter did not specifically examine geriatric nurses.

Regarding the compassion satisfaction variable, there was no significant change when comparing the two groups, though there was a significant change at the follow-up of the experimental group at three months post-intervention. The lack of a significant initial change could be because the study sample reported average to high satisfaction at the beginning of the study, which is relevant in this professional group, especially when they have been doing this work for years [58,59].

Dementia tends to progress slowly, and the average time from the onset of overt symptoms to death is about 8–10 years. Symptoms usually develop slowly and worsen over time, and, therefore, the demands on the daily caregivers of people with dementia increase progressively [60]. Therefore, interventions tailored to healthcare professionals should be sustainable, while empowering them to cope with daily challenges [5]. Our results showed that the beneficial effect of the combined program based on MBSR and MBCT could last at least three months after the intervention. This finding may be related to the benefits of the regular practice of mindfulness among nurses [33,61]. This program provided five audio recordings (MP3) of guided mindfulness exercises for the nurses in the experimental group to use as practice, plus two assignments per week, and we also monitored their progress by email to encourage the nurses to cultivate the habit of practicing mindfulness and to apply a mindful attitude to their daily activities. We believe that this is the main reason for the sustainable effects we found in our study, supported by the systematic reviews conducted on the influence of mindfulness training [34,61].

Understanding the effects of caring for institutionalized patients with dementia on nurses is a responsibility of the institution. Although concepts such as compassion fatigue and burnout are multifactorial [62], studies suggest that the social environment of a workplace and its organizational structure are particularly relevant contributors to these conditions [16,62]. The results of this study suggest the need for an intervention for at-risk staff, since modifying the organizational structure is often more complex. If we provide professionals with personal tools that favor their self-care and emotional self-regulation, we will contribute to improving their health, and thus the quality of their work [52,63]. This will have a relevant impact on the care of the older people they attend, supported by multiple studies [64,65].

Similar studies, as well as systematic reviews, point to the effectiveness of mindfulness training-based interventions (either for stress reduction or within cognitive therapy, combined in our study) for enabling caregivers to self-manage their stress under adverse work conditions with a high care overload [66,67]. These results are of great importance because they suggest that decreasing compassion fatigue and burnout may prevent emotional problems and symptomatology [36,68]. The longitudinal study design is also relevant, because the development of brief, online interventions that are easy to implement and follow, with sustained long-term effects that promote training, is highly applicable [69]. Further, the online nature of the training, which favors self-paced training, makes it highly replicable. Thus, organizations with these characteristics can implement this approach to prevent or reduce the levels of compassion fatigue and burnout of their professionals [13,70,71].

However, although the mindfulness-based interventions developed over the last 30 years are increasingly used by healthcare professionals to reduce the risk of burnout, they have had varying results [55]. A recent review of the literature on mindfulness-based interventions for stress reduction in professionals concluded that there is still a lack of evidence regarding the effectiveness of these interventions [36]. A possible explanation may be the heterogeneity of the numerous intervention types (approach, content, methodology, and duration) [18]. Another recent systematic review of 44 meta-analyses of RCTs clearly suggested that more rigorous studies are needed, using randomized and controlled trials that highlight longitudinal effects, as is the case in our study [34].

Therefore, three relevant aspects of our study are worth highlighting: (1) this was a RCT with a control group, meaning that the variables have been rigorously controlled and a random selection has been made; (2) the intervention was online, which can be replicated and implemented at a low cost, which demonstrates its effectiveness; (3) it has shown a longitudinal benefit according to the regression model, with a six-week intervention lasting 120 min, which, compared to the literature, makes it a short intervention, which also enhances its applicability and effectiveness. The advantage of using online training in this environment is that staff can access the course when convenient and at their own pace, and they can immediately use the techniques learned in their practice.

Although our findings, despite being a RCT, must be taken with caution, and given the heterogeneity of mindfulness-based interventions and the lack of independent research groups replicating specific mindfulness-based interventions, research in this field clearly indicates that engaging in mindfulness-based practices helps healthcare professionals improve their well-being.

Despite the usefulness of our findings, this study is not exempt from limitations. We cannot generalize these findings to other settings such as the public sector, which would be interesting, since working conditions and professional ratios may have an influence. Another limitation is that, despite the difficulties of access to change and the fact that it is an RCT, the sample (74 nurses) is too small to draw generalizable conclusions. Another limitation—although in this study, it is seen as an advantage—is the fact that the intervention was not conducted face-to-face, but via an online format. Several studies suggest that this approach makes it less effective, since in the literature, training tends to be face-to-face and led by an instructor.

## 5. Conclusions

Future research could replicate this intervention in a sample of professionals from public institutions and caregivers with high compassion fatigue, high burnout, and low satisfaction, as well as in larger samples and in different sectors to determine whether these findings can be generalized. It would also be worthwhile to test the intervention with family caregivers outside the scope of institutionalized care. Further research is needed on online mindfulness training and interventions to improve the biopsychosocial health of nurses during the COVID-19 pandemic. If it is known that these interventions are effective when applied to work environments with high fatigue and stress load, it would be worthwhile to replicate them during this pandemic and post-pandemic era.

## Figures and Tables

**Figure 1 ijerph-19-11441-f001:**
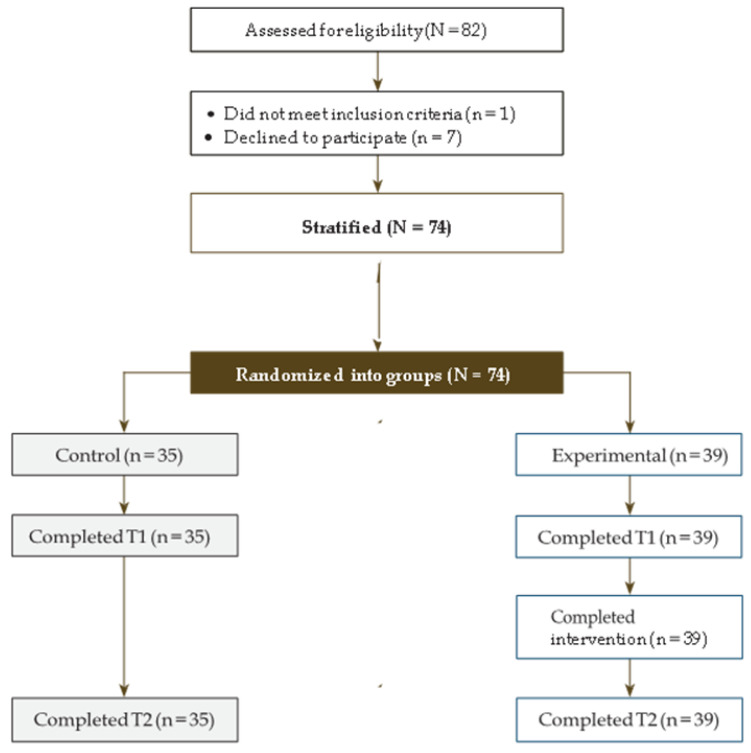
CONSORT trial participation at baseline (T0) and three months (T2).

**Table 1 ijerph-19-11441-t001:** Subscales of the ProQoL questionnaire at baseline for the control and intervention groups.

Variables	Experimental (*n* = 39)		Control (*n* = 35)		*p* *
	M (SD)	Range	M (SD)	Range	
Professional Quality of Life ProQoL (0–50)					
Satisfaction	40.49 (7.22)	29–50	41.25 (5.02)	11–50	0.087
Compassion fatigue	16.44 (4.45)	2–30	17.45 (7.12)	2–47	0.065
Burnout	17.88 (7.33)	8–28	18.38 (4.05)	4–34	0.074

* Student’s *t* test (significance *p* < 0.05).

**Table 2 ijerph-19-11441-t002:** Summary of multiple linear regression analyses predicting subject variables, including the three subscales (ProQOL). Controlling for the effect of comparing the experimental group with the control group after controlling for time.

Model	R^2^	R^2^ Change	Variable	Standardized β	*p*
ProQOL: Compassion satisfaction					
1	0.45	0.45			<0.001
			Time	0.62	<0.001
2	0.50	0.11			<0.001
			Exp. vs. control	0.31	<0.001
ProQOL: Compassion fatigue					
1	0.40	0.40			<0.001
			Time	−0.61	<0.001
2	0.61	0.16			<0.001
			Exp. vs. control	−0.69	<0.001
ProQOL: Burnout					
1	0.38	0.37			<0.001
			Time	−0.60	<0.001
2	0.44	0.11			<0.001
			Exp. vs. control	−0.32	<0.001

## Data Availability

Not applicable.

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
