# Peer review of "Mindfulness-Based Intervention for the Reduction of Compassion Fatigue and Burnout in Nurse Caregivers of Institutionalized Older Persons with Dementia: A Randomized Controlled Trial"

_ijerph, 2022, doi:10.3390/ijerph191811441_

Round 1

Reviewer 1 Report

Dear Authors, 

Greetings to you all. The great attempt to study, Mindfulness‐Based Intervention for the Reduction of Compassion Fatigue and Burnout in Nurse Caregivers of Institutionalized Older Persons with Dementia: A Randomized Controlled Trial. The study is narrated in a better way to understand the real scenario of the research focus. Few things can be done to enhance the paper quality. 

1. Abstract can be rewritten for better clarity. 

2. Results were discussed only in the tables visually can be presented. 

3. Is this sample size enough for the study? 

4. What theory was adopted for this study? if any kindly discuss them in the literature summary. 

5. Conclusions need improvements; inferences of the results can be added to them. Directly the first line says about the future scope. this should be discussed last part of the conclusions. 

Author Response

Thank you for taking the time to review our manuscript. We are very grateful for the reviewers’ comments on our paper. We have considered them with care, and the comments have been valuable for us when improving the manuscript. Please find below the comments, our responses and changes made.

Reviewer 2 Report

The research contributes to the field by showing a direct evaluation of the efficacy of a combined online training in two types of mindfulness‐based therapy for the reduction of compassion fatigue and burnout in geriatric nurses caring for institutionalized older people with dementia in a randomized controlled trial. However, the positive psychological effects of mindfulness in nurses have been explained by many researchers with advanced statistical methods. In addition, there is no research design and statistical evaluation method suitable for the purpose of the research. In this case, we find that the manuscript does not present sufficiently original or distinct findings that are of relevance to a regional or international audience.

Author Response

(The authors gave the same response as above.)
